# 3D Reconstruction and Novel View Synthesis of Indoor Environments based on a Dual Neural Radiance Field

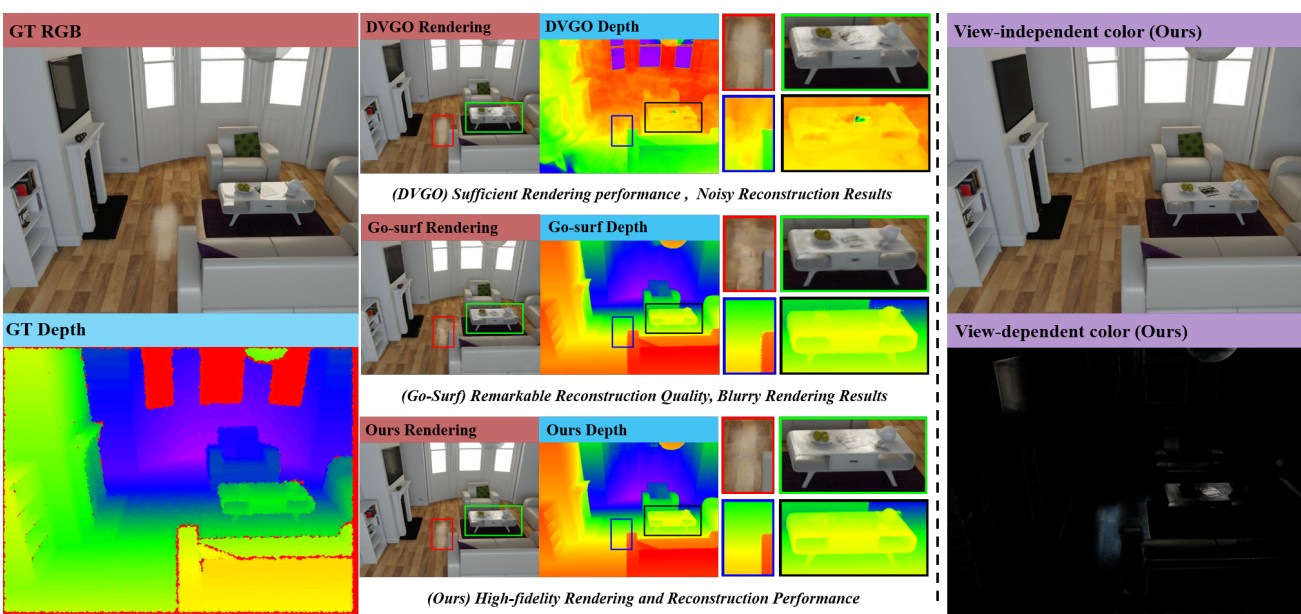

(a) Reconstruction and Rendering results

(b) Color Decomposition results

**Figure 1: NeRF-based view synthesis methods (eg. *DVGO*) often produce noisy 3D reconstruction results (top right in fig. 1 (a)), while NeRF-based reconstruction approaches (eg. *Go-Surf*) struggle to produce satisfactory rendering image (middle right in fig. 1 (a)). We propose a dual-field architecture to simultaneously obtain high-fidelity view synthesis and reconstruction performance (bottom right in fig. 1 (a)). Fig. 1 (b) illustrates the view-independent and view-dependent color components disentangled by our proposed method.**

## ABSTRACT

Simultaneously achieving 3D reconstruction and novel view synthesis for indoor environments has widespread applications but is technically very challenging. State-of-the-art methods based on implicit neural functions can achieve excellent 3D reconstruction results, but their performances on new view synthesis can be unsatisfactory. The exciting development of neural radiance field (NeRF) has revolutionized novel view synthesis, however, NeRF-based models can fail to reconstruct clean geometric surfaces. We have developed a dual neural radiance field (Du-NeRF) to simultaneously achieve high-quality geometry reconstruction and view rendering. Du-NeRF contains two geometric fields, one derived from the SDF field to facilitate geometric reconstruction and the other derived from the density field to boost new view synthesis. One of the innovative features of Du-NeRF is that it decouples a view-independent component from the density field and uses it as a label to supervise the learning process of the SDF field. This reduces shape-radiance ambiguity and enables geometry and color to benefit from each other during the learning process. Extensive experiments demonstrate that Du-NeRF can significantly improve the performance of novel view synthesis and 3D reconstruction for indoor environments and it is particularly effective in constructing areas containing fine geometries that do not obey multi-view color consistency.

*ACM MM, 2024, Melbourne, Australia*
© 2024 Copyright held by the owner/author(s). Publication rights licensed to ACM.
ACM ISBN 978-x-xxxx-xxxx-x/YY/MM
https://doi.org/10.1145/nnnnnnn.nnnnnnn

**Unpublished working draft. Not for distribution.**

## CCS CONCEPTS

• **Computing methodologies → Computer vision**; **Computer graphics**.

## KEYWORDS

Reconstruction, Rendering, Neural radiance field, Indoor scene.

## 1 INTRODUCTION

Novel view synthesis and 3D reconstruction for indoor environments are of great interest in the computer vision and graphics communities [1, 8, 11, 13, 15, 16, 23, 24, 46, 50]. They provide fundamental support for applications such as robot perception and navigation, virtual reality, and indoor design. Classical indoor 3D reconstruction methods perform registration and fusion to obtain dense geometry using depth images in an explicit manner where the depth images are obtained either with range sensors like Kinect or inferred from RGB images [5, 7, 10, 12, 14, 25, 34, 43]. However, due to noise and holes in the depth images, complete and smooth indoor geometry is difficult to generate. Additionally, such an explicit representation can often fail to preserve sufficient details due to storage limitations thus making it very difficult to synthesize realistic novel views.

In recent years, coordinate neural networks have been extensively used to describe 3D geometry and appearance due to their powerful implicit and continuous representation capacities [22, 23, 28, 29, 38, 40, 44]. These models take the 3D coordinates as input and output the signed distance value [28, 40, 44], density [23, 24], or occupancy of the scene [22, 29, 38]. Although methods such as *Neus* [40] and *VolSDF* [44] can achieve accurate 3D reconstruction of objects, they do not perform well in indoor scenes containing textureless regions or when the observations are sparse. In such cases, depth images are often introduced to provide additional supervision for network training to improve performances [1, 39, 46]. Furthermore, these methods focus on 3D reconstruction and their performances on novel view synthesis can be unsatisfactory (middle right in fig. 1(a)).

Neural Radiance Field (NeRF) [23] and a series of its extensions [2–4, 6, 9, 15, 17, 20, 24, 36, 41, 45, 47–49] have achieved exciting results in novel view synthesis. It implicitly represents the density field and color field and performs novel view synthesis via volume rendering. However, these methods can fail to reconstruct clean indoor surfaces (top right in fig. 1(a)), as the density used in NeRF samples the whole space rather than in the vicinity of the surfaces.

In this paper, we propose a dual neural radiance field (Du-NeRF) to simultaneously achieve high-quality geometry reconstruction and view rendering (bottom right in fig. 1(a)). Specifically, our framework contains two geometric fields, one is derived from the SDF field with clear boundary definitions, and the other is a density field that is more conducive to rendering. They share the same underlying input features, which are interpolated from multi-resolution feature grids and then decoded by different decoders. We enable the two branches to each play their respective strength, while the former is used to extract geometric features, the latter is used to support the task of new view synthesis. In addition, we decouple a view-independent component from the density field and use it as a label to supervise the learning process of SDF during the network optimization process (the top row in fig. 1(b)). In our method, the two geometric fields share the underlying input geometric features to facilitate the optimization of the underlying geometric feature grid. Moreover, we use a view-invariant component decoupled from

the density branch to replace the view-varying ground truth (GT) images to guide the geometric learning process to reduce shape-radiance ambiguity and allow geometry and color to benefit from each other during the learning process. Experimental results show that this design can effectively construct fine geometries to achieve smooth scene reconstruction, especially in those areas that do not obey multi-view color consistency. Our contributions are as follows:

- We have developed a novel neural radiance field termed Dual Neural Radiance Field (Du-NeRF) for simultaneously improving 3D reconstruction and new view synthesis of indoor environments.
- We introduce a novel self-supervised method to extract a view-independent color component for supervising 3D reconstruction, which significantly enhances the smoothness of surfaces and fills in missing parts of indoor objects.
- Extensive experiments demonstrate that our method can significantly improve the performance of novel view synthesis and 3D reconstruction for indoor environments.

## 2 RELATED WORK

**Neural radiance-based novel view synthesis.** The introduction of the neural radiance field (NeRF) marks remarkable progress in novel view generation. NeRF models a full-space implicit differentiable and continuous radiance field with neural networks and uses volume rendering to obtain color information. Many variants have been proposed to improve training, inferencing and rendering performances [2–4, 6, 17, 20, 24, 36, 41, 48]. In particular, *DirectVoxelGO* [36] and *InstantNGP* [24] combine explicit and implicit representations and use hybrid grid representations and shallow neural networks for density and color estimation respectively to achieve faster rendering and higher rendering quality. Liu *et al.* [20] introduce a progressive voxel pruning and growing strategy to sample the effective region near the scene surface. Chen *et al.* [4] use a combination of 2D planes and 1D lines to approximate the grid to achieve faster reconstruction, improved rendering quality, and smaller model sizes. To address the ambiguity arising from pixels represented by a single ray, Barron *et al.* [2] introduce the concept of cones instead of points, to increase the receptive field of a single ray. This approach, known as *Mip-nerf*, effectively tackles issues of jaggies and aliasing and enhances rendering quality. To reduce the blurring effect, Lee *et al.* [17] design a rigid blurry kernel module which takes into account both motion blur and defocus blur during the real acquisition process. Kun *et al.* [48] further improve the rendering performance by learning a degradation-driven inter-viewpoint mixer. Additionally, some works attempt to improve the rendering performance by jointly optimizing the poses of the training images [3, 6, 41]. In contrast, our method adopts geometry-guided sampling, which benefits from the reconstruction result and allows for more accurate sampling of points near the surface, thus improving the performance of view synthesis.

**Neural implicit 3D reconstruction.** Neural implicit functions take a 3D location as input and output occupancy, density, and color [22, 23, 26, 27, 29]. *Scene Representation Networks* employ MLPs to map 3D coordinates to latent features that encode geometry and color information [35]. Yariv *et al.* [44] and Wang *et al.* [40] propose two approaches to converting SDF values into density and

performing volume rendering to supervise object reconstruction in the Nerf-based framework. *Neuralangelo* [19] introduces numerical gradients and utilizes multiresolution hash grids to reconstruct detailed scenes. However, while these methods perform well on scenes with rich texture, they struggle with indoor scenes with textureless walls and ceilings. To address these issues, Yu *et al.* [46] employ predicted depth and normal maps from a pre-trained network to enhance the reconstruction of indoor scenes. Azinović *et al.* [1] combine a TSDF representation with the NeRF framework and use a depth representation from an off-the-shelf RGBD sensor to improve the accuracy of indoor geometry reconstruction. Subsequent studies [18, 39, 42] have further optimized the Neural-RGBD strategy to speed up training. [39] utilizes voxel representations instead of 3D coordinates to achieve faster query, while [42] trains a dynamically adaptive grid that allocates more voxel resources to more complex objects. [18] pre-trains a feature grid to accelerate the training process. We propose to decouple the view-independent colour component for guiding the 3D reconstruction, which effectively enhances the smoothness of surfaces and fills in the missing parts of depth-based reconstruction.

## 3 METHOD

### 3.1 Preliminaries

**Neural radiance field.** Given a collection of posed images, neural radiance field can estimate the color and depth of each pixel by computing the weighted sum of sampling points based on their color and distance from the center of the camera [21]:

$$C_p = \sum_{i=1}^{N} w_i c_i, D_p = \sum_{i=1}^{N} w_i z_i, \tag{1}$$

where the color and distance from sampling points to the camera center, are denoted by $c_i$ and $z_i$ respectively, $w_i$ are the contribution weights of each sampling point to the color and depth and is calculated as $w_i = \sum_{i}^{N}(\prod_{j}^{i-1}(1 - \alpha_j(p(x))))\alpha_i(p(x_i))$, where $p(x)$ and $\alpha_i$ are the density and opacity of sampling point $x_i$, respectively. The opacity is then computed using eq. (2).

$$\alpha_i = 1 - exp(-p(x_i)(z_{i+1} - z_i)), \tag{2}$$

the color $c_i$ and density $p(x_i)$ of a given sampling point are predicted by the Multi-layer Perceptron (MLP). The process is illustrated in eq. (3).

$$\begin{aligned} p(x_i), f &= \Gamma_\theta(x_i), \\ c_i &= \Gamma_\kappa(f, d), \end{aligned} \tag{3}$$

where $x$ and $d$ represent the location and ray direction, respectively. $f$ is the feature vector related to the location. $\Gamma_\theta$ and $\Gamma_\kappa$ are implicit functions modelled by MLPs.

**SDF-based neural implicit reconstruction.** The signed distance function (SDF) refers to the nearest distance between a point and surfaces and is often used to implicitly represent geometry. One notable application of SDF is neural implicit reconstruction under the volume rendering framework, achieved by a method called *Neus* [40].

The key to the success of this method is an unbiased transformation between the SDF values and the density, as demonstrated in eq. (4). This transformation enables the creation of high-quality surface geometry with great accuracy and cleanliness.

$$\alpha_i = \max\left(\frac{\sigma_s\left(\phi\left(\mathbf{x}_i\right)\right) - \sigma_s\left(\phi\left(\mathbf{x}_{i+1}\right)\right)}{\sigma_s\left(\phi\left(\mathbf{x}_i\right)\right)}, 0\right), \tag{4}$$

where $\phi(x_i)$ represents the SDF value of a given sample point, and $\sigma_s(x) = (1 + e^{-sx})^{-1}$, while the smoothness of the surface is conditioned on a learnable parameter $s$.

**Multi-resolution feature grid.** To improve the efficiency of training, a grid representation is used [4, 36, 45]. However, using only single-resolution grid limits the optimization of density and color to local information, resulting in disruptions to the smoothness and continuity of the scene texture [39, 46]. A multi-resolution grid expands the local optimization to nearby continuous fields by varying the receptive field and gradient backpropagation of sample points, enabling higher rendering quality and smoother geometry. The embedding feature is obtained by concatenating the features of each level, as shown in eq. (5).

$$f = \Omega(V_1(x), V_2(x), ..., V_n(x)), \tag{5}$$

where $\Omega$ indicates concatenation, and $V_i(x)$ is trilinear interpolation in the $i$-th grid. The final feature vector of the input network is denoted as $f$.

To achieve a higher resolution, hash-based feature grid is employed in [24]. It represents resolutions as:

$$R_l := \left\lfloor R_{\min} b^l \right\rfloor, \ b := \exp(\frac{\ln R_{\max} - \ln R_{\min}}{L - 1}), \tag{6}$$

where $R_{\min}$, $R_{\max}$ are the coarsest and finest resolution, respectively. $R_l$ represents $l-$th level resolution and $L$ is the total levels. Similarly, we extract the interpolated features at each level and concatenate them together as in eq. (5).

### 3.2 Dual Neural Radiance Field

**Scene representation.** To achieve high-fidelity indoor scene reconstruction and rendering simultaneously, we introduce the dual neural radiance field (*Du-NeRF*). The key idea that enables the dual neural radiance field to achieve high-fidelity reconstruction and rendering is to separately represent the geometry field and the color field. Taking into account the fact that the reconstruction task and the rendering task have different complexities, the geometry field and the color field are represented by multi-resolution grids with different levels to speed up training and inference.

As shown in fig. 2, we use a four-level grid to represent the scene geometry where the grid sizes at each level are respectively 3cm, 6cm, 24cm, and 96cm. At each level, the dimension of the geometry feature is set to 4, and the dimension of the geometry feature $f_{gi}$ is therefore 16. For the color field, the hash-based multi-resolution grid is utilized. The coarsest resolution $R_{\min}$ of the hash-based multi-resolution grid is set to 16. The level of grid resolution is $L = 16$, and the feature vector at each level is 2-dimensional, we therefore obtain a hash color feature vector $f_{ci}$ with a total of 32 dimensions. The geometry feature $f_{gi}$ and color feature $f_{ci}$ are obtained using tri-linear interpolation.

**Dual neural radiance networks.** The key idea is to use two different geometric decoders to extract SDF and density, which are respectively used for 3D reconstruction and image rendering. The

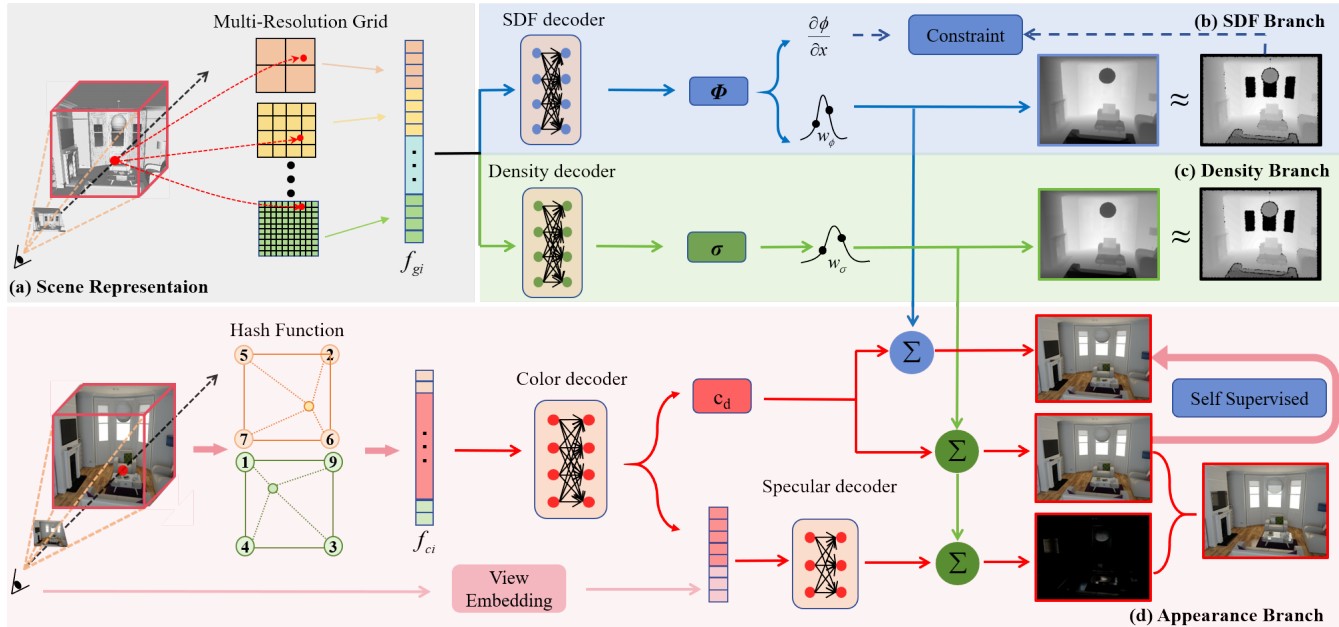

**Figure 2: Dual Neural Radiance Field (Du-NeRF).** In (a) Scene Representation, we use a four-layer multi-resolution grid to store geometric features $f_{gi}$, and a hash-based multi-resolution grid for the color features $f_{ci}$. $f_{gi}$ is decoded to SDF $\phi$ and density $\sigma$ by different MLPs in (b) SDF branch and (c) the Density branch. We provide depth constraints for $\phi$ and introduce additional regularization terms to ensure the stability of its training. We design a depth alignment loss for $\sigma$ to align the two geometry fields. In (d), for color calculation, $\phi$ and $\sigma$ from (b) and (c) are integrated with decoupled view-independent colors to compute the self-supervised loss. The final rendering color sums the view-dependent and view-independent colors to be integrated with the $\sigma$ weights.

SDF and density are estimated implicitly from the same interpolated features $f_{gi}$:

$$\Gamma_\phi(f_{gi}) = \phi_i, \ \Gamma_\sigma(f_{gi}) = \sigma_i, \tag{7}$$

where both of the $\Gamma_\phi$ and $\Gamma_\sigma$ are MLPs, and used to decode $f_{gi}$ into SDF $\phi_i$ and density $\sigma_i$, respectively. $\phi_i$ and $\sigma_i$ are both converted to occupancy through eq. (4) and eq. (2), respectively. The resulting occupancy of the two is calculated via eq. (1) to obtain the pixel depth and color. The whole process is supervised with the image reconstruction loss and depth loss.

Sampling points near object surface have a higher contribution to the rendered color of the ray [23, 40]. To obtain higher rendering quality, we employ the hierarchical sampling strategy as in [23, 40], which contains coarse sampling and fine sampling near the surface. Specifically, we first uniformly sample 96 points along the ray in the coarse stage and then iteratively add 12 sampling points three times according to the cumulative distribution function(CDF) of previous coarse points weights in the fine stage as in [39]. Finally, we got 132 sampling points for depth and color rendering. Note that we use the weight distribution calculated from the SDF branch for sampling as it provides more accurate surface information. For the volume rendering process, the two branches share the sampling points.

**Self-supervised color decomposition.** We disentangle the color into view-independent color and view-dependent color. We use the decoupled view-independent color $c_{di}$ to guide geometry

learning in a multi-view consistent self-supervised manner by constraining the weight values. This separation allows the color branch to leverage complete color information for rendering, while the geometry branch benefits from view-consistent supervision. Previous work has shown that decoupling color helps find geometry surfaces in NeRF-based methods [37, 49]. For instance, [37] accomplishes the color decomposition via a simple regularization term, which assumes that the view-dependent color is close to zero. In contrast, we utilize the view-independent color decoupled from the two branches to constrain each other in a self-supervised manner. This design effectively extracts view-independent color to support accurate geometry reconstruction but also boosts the image rendering results via the mutually beneficial learning process.

To achieve it, as shown in fig. 2, we utilize two color decoders consisting of MLPs to implicitly estimate the view-independent color and view-dependent color. The view-independent decoder takes the interpolated color feature $f_{ci}$ as input and outputs the view-independent color $c_{di}$ and an intermediate feature $f_{ini}$. The specular (view-dependent) decoder takes the intermediate feature $f_{ini}$ and the encoded view-direction vectors as input to obtain the specular color $c_{si}$. The overall color at a sampling point is calculated by adding the two color components $c_i = c_{di} + c_{si}$. The final color $C$ is generated by summing the weighted color of each sampling point with eq. (1). In our framework, we integrate $\omega_{\sigma_i}$ with the full color $c_i$ as in *NeRF*, however, we only weight the view-independent

color $c_{di}$ with $\omega_{\phi_i}$ to obtain the view-independent color of the corresponding ray. The calculation of color in eq. (1) becomes:

$$C_{d\phi} = \sum_{i=1}^{N} \omega_{\phi i} c_{di}, \tag{8}$$

where $C_{d\phi}$ is the view-independent color computed by $\omega_{\phi i}$. We can obtain the view-independent color $C_{d\sigma}$ computed by $c_{di}$ and $\omega_{\sigma_i}$, which will be used as the ground truth to constrain the learning process of $C_{d\phi}$:

$$\mathcal{L}_d = \sum_{i}^{N} \lambda_d \left\| C_{d\phi} - C_{d\sigma} \right\|. \tag{9}$$

## 3.3 Pose Refinement

Noisy poses lead to undesirable mesh protrusions of 3D reconstruction and artifact in novel view synthesis. Thus, we adopt a pose optimization strategy in the training process, which treats the poses as learnable parameters as in [1]. Specifically, we convert the transformation matrix to Euler angles and translation vectors ($\mathbb{R}^{3+3}$) and initialize them with results of the Bundlefusion[8]. We optimize them along with the dual fields and are supervised by depth loss and photometric loss.

After pose optimization, poses of training images are transformed into a new coordinate space. A calibration process is required for testing images to transform them into the same coordinate space as the training images. To achieve it, we introduce the proximity frame alignment (PFA) strategy, which uses the pose change of the adjacent training images to help correct the noisy poses of the testing images. Specially, given a sequence of consecutive images $\{I_i \in \mathbb{R}^{H \times W \times 3} \mid i \in \{1, ...N\}\}$, and their corresponding poses $\{P_i \in \mathbb{R}^{4 \times 4} \mid i \in \{1, ...N\}\}$, we suppose that the $k$−th image is testing image and the $k+1$-th is the training image. $P_k$ and $P_{k+1}$ are noisy poses, and $P'_{k+1}$ is the optimised pose of the training image, respectively. $P'_k$ is the correct pose of the testing image, which can be calculated by eq. (10).

$$P'_k = \mathfrak{A} \times P_k, \tag{10}$$

where $\mathfrak{A}$ is the transformation matrix of the adjacent training image $k+1$, obtained by:

$$\mathfrak{A} = P'_{k+1} \times P_{k+1}^{-1}. \tag{11}$$

In this way, we can calibrate the testing images into the coordinate system of the training images.

## 3.4 Network training

To optimize the dual neural radiance field, we randomly sample $M$ rays during training. Our loss is divided into two components including a $\mathcal{L}_\phi$ loss, and a $\mathcal{L}_\sigma$ loss in eq. (12).

$$\mathcal{L}(\mathrm{P}) = \mathcal{L}_\phi + \mathcal{L}_\sigma. \tag{12}$$

The $\mathcal{L}_\phi$ contains the three components as shown in eq. (13): view-independent color loss, depth loss and SDF regularization loss, as the following:

$$\mathcal{L}_\phi = \lambda_d \mathcal{L}_d + \lambda_{\mathrm{depth}} \mathcal{L}_{\mathrm{depth}} + \mathcal{L}_{\mathrm{SDF}}. \tag{13}$$

The view-independent loss is calculated as the distance between $C_{d\phi}$ and $C_{d\sigma}$ as shown in eq. (9), and the depth loss is the $L_1$ loss:

$$\mathcal{L}_{\mathrm{depth}} = \sum_{i}^{N} \left| D_\phi - D_{\mathrm{gt}} \right|. \tag{14}$$

In order to improve the robustness of the learning process of the SDF $\phi$, we imposed a series of SDF losses $\mathcal{L}_{\mathrm{SDF}}$ to regularize the $\phi$ value as [39]:

$$\mathcal{L}_{\mathrm{SDF}} = \lambda_{\mathrm{eik}} \, \mathcal{L}_{\mathrm{eik}} \, + \lambda_{\mathrm{fs}} \mathcal{L}_{\mathrm{fs}} + \lambda_{\mathrm{sdf}} \, \mathcal{L}_{\mathrm{sdf}} \, + \lambda_{\mathrm{smooth}} \, \mathcal{L}_{\mathrm{smooth}} \,. \tag{15}$$

The regularization term $\mathcal{L}_{\mathrm{eik}}(x)$ encourages valid SDF predictions in the unsupervised regions, while $\mathcal{L}_{\mathrm{smooth}}(x)$ is an explicit smoothness term to realize smooth surfaces.

$$\mathcal{L}_{\mathrm{eik}}(x) = (1 - \|\nabla\phi(x)\|)^2, \tag{16}$$

$$\mathcal{L}_{\mathrm{smooth}}(x) = \|\nabla\phi(x) - \nabla\phi(x+\epsilon)\|^2. \tag{17}$$

$\mathcal{L}_{\mathrm{sdf}}(x)$ and $\mathcal{L}_{\mathrm{fs}}(x)$ are used to constrain the truncation distance $b(x)$.

$$\mathcal{L}_{\mathrm{sdf}}(x) = |\phi(x) - b(x)|, \tag{18}$$

$$\mathcal{L}_{\mathrm{fs}}(x) = \max \left( 0, e^{-\alpha\phi(x)} - 1, \phi(x) - b(x) \right). \tag{19}$$

A detailed explanation of these regularity constraints can be found in [39].

The $\mathcal{L}_\sigma$ loss contains two parts as shown in eq. (20):

$$\mathcal{L}_\sigma = \lambda_{\mathrm{rgb}} \sum_{i}^{N} \left\| C_\sigma - C_{\mathrm{gt}} \right\| + \lambda_{\mathrm{align}} \sum_{i}^{N} \left| D_\sigma - D_{\mathrm{gt}} \right|, \tag{20}$$

where $C_\sigma$ is the final rendering color and $D_\sigma$ is the depth calculated from the color branch. $\lambda_{\mathrm{align}}$ is used to align two geometric fields. We experimentally found that $\mathcal{L}_{\mathrm{align}}$ is important for decoupling consistent view-independent color. For each of the geometry coefficients, we follow the practice of [39] and set them as follows: $\lambda_d = 5$, $\lambda_{\mathrm{depth}} = 1$, $\lambda_{\mathrm{eik}} = 1.0$, $\lambda_{\mathrm{fs}} = 1.0$, $\lambda_{\mathrm{sdf}} = 10.0$, $\lambda_{\mathrm{smooth}} = 1.0$. For color coefficients we experimentally set them as $\lambda_{\mathrm{rgb}} = 50$ and $\lambda_{\mathrm{align}} = 1$.

# 4 EXPERIMENT

## 4.1 Setup

**Datasets.** We utilize the *NeuralRGBD* (10 scenes), *Replica* (8 scenes), and *Scannet* (5 scenes) datasets to evaluate the proposed method. The *NeuralRGBD* dataset and *Replica* dataset are synthetic datasets, and images are of high quality. The *Scannet* dataset is real world dataset, captured with a handheld device such as an iPad and suffers from severe motion blur. Each scene in the three datasets includes RGB images, depth images, and corresponding poses. The poses are obtained via the *BundleFusion* algorithm. For each scene, about 10 percent of images are used as validation sets, and the rest of the images are used as training sets. Specifically, starting with the 10-th image, we choose one image in every ten consecutive images as the validation set.

**Implementation details.** The SDF decoder $\Gamma_\phi$, density decoder $\Gamma_\sigma$, color decoder $\Gamma_d$ and specular decoder $\Gamma_s$ all use two-layers MLPs with the hidden dimension of 32. We sample $M = 6144$ rays for each iteration, and each ray contains $N_c = 96$ coarse samples and $N_f = 36$ fine samples. Our method is implemented in Pytorch and trained with the ADAM optimizer with a learning rate of $1 \times 10^{-3}$,

and $1 \times 10^{-2}$ for MLP decoders, multi-resolution grids features, respectively. We run 20K iterations in all scenes with the learning rate decay at iteration 10000 and 15000, and the decay rate is 1/3.

**Baselines.** To validate the effectiveness of the proposed method, we compare it with approaches for indoor 3D reconstruction and view rendering, respectively. For indoor 3D reconstruction, we compare with registration-based method including *BundleFusion* [8], *Colmap* [30–32], *Convolutional Occupancy Networks* [29], *SIREN* [33], and recent volumetric rendering-based geometry reconstruction methods such as *Neus* [40], *VolSDF* [44], *Neuralangelo* [19], *Neural RGBD* [1], and *Go-Surf* [39]. We run marching cubes at the resolution of 1cm to extract meshes. We cull the points and faces in the areas that are not observed in any camera views as in [1, 39]. For view rendering, we compare some representative approaches based on the neural radiance field, including *DVGO* [36], and *InstantNGP* [24]. Their performance on 3D reconstruction is also compared.

## 4.2 Comparison

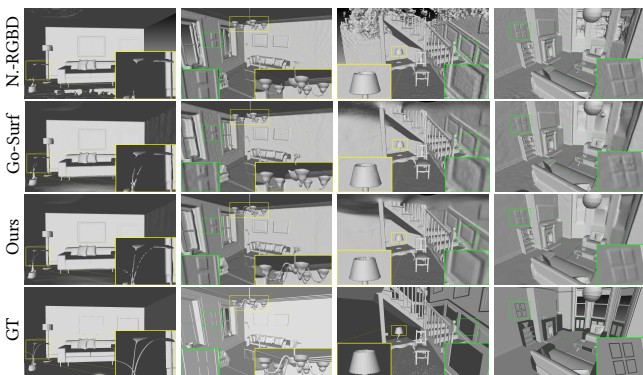

Figure 3: The qualitative reconstruction results on Neural-RGBD datasets. The proposed method can fill in the missing part (highlighted in yellow boxes) and produce smoother planes and clear edges (highlighted in green boxes)

**Evaluation on NeuralRGBD Dataset.** We evaluated our method on 10 synthesis data used in *Neural RGB-D* and *Go-Surf*. Qualitative and quantitative results are shown in figs. 3 and 4 and table 1. It can be seen from the table 1 that our method achieves the best performance on both view rendering and 3D reconstruction. Detailed rendering and reconstruction results can be found in the Supplementary.

For the 3D reconstruction task, the proposed method shows superior results on all metrics except the *NeuralRGBD* approach in terms of accuracy (table 1). fig. 3 shows that our method generate smoother geometry ( green boxes), and can fill in the missing stripes of the indoor objects (yellow boxes). For the view synthesis task, our method shows an improvement of $3 - 10$ db over the previous representative approaches (table 1), and the rendered images have richer textures (fig. 3), *e.g.* the subtitles on the TV and the magazine on the table.

**Evaluation on Replica dataset.** As shown in table 2, our method also demonstrates the superiority on the Replica dataset.

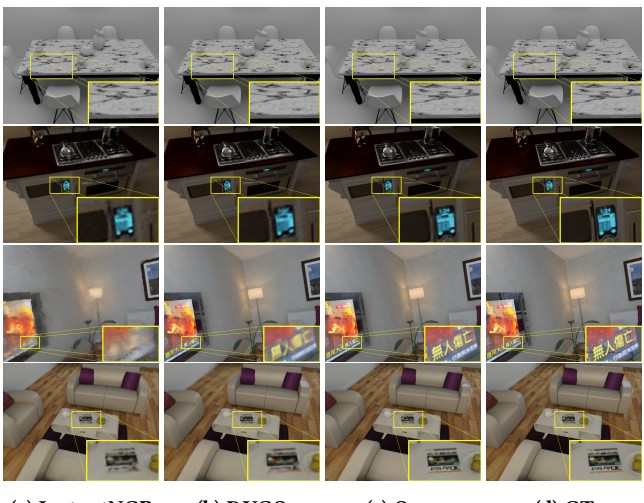

| (a) InstantNGP | (b) DVGO | (c) Ours | (d) GT |

Figure 4: Qualitative comparison of novel view synthesis results of NeuralRGBD dataset. It can be seen from the results that our method has better visual rendering effects, whether they are striped structures or text on books.

Table 1: Reconstruction and view synthesis results of NeuralRGBD dataset. The best performances are highlighted in bold.

| Method | Acc ↓ | Com ↓ | C-$l_1$ ↓ | NC ↑ | F-score ↑ | PSNR ↑ | SSIM ↑ | LPIPS ↓ |
|---|---|---|---|---|---|---|---|---|
| BundleFusion | 0.0178 | 0.4577 | 0.2378 | 0.851 | 0.680 | - | - | - |
| COLMAP | 0.0271 | 0.0364 | 0.0293 | 0.888 | 0.874 | - | - | - |
| ConvOccNets | 0.0498 | 0.0524 | 0.0511 | 0.861 | 0.682 | - | - | - |
| SIREN | 0.0229 | 0.0412 | 0.0320 | 0.905 | 0.852 | - | - | - |
| Neus | 0.3174 | 0.6911 | 0.5043 | 0.628 | 0.103 | 27.465 | 0.849 | 0.192 |
| VolSDF | 0.1627 | 0.4815 | 0.3222 | 0.681 | 0.262 | 28.717 | 0.882 | 0.174 |
| Neuralangelo | 0.3212 | 0.6938 | 0.5075 | 0.531 | 0.101 | 29.786 | 0.892 | 0.121 |
| Neural RGB-D | **0.0145** | 0.0508 | 0.0327 | 0.920 | 0.936 | 31.994 | 0.901 | 0.183 |
| Go-Surf | 0.0164 | 0.0213 | 0.0189 | 0.932 | 0.949 | 29.586 | 0.889 | 0.183 |
| DVGO | 0.2389 | 0.5558 | 0.3973 | 0.564 | 0.317 | 33.633 | 0.940 | 0.125 |
| Instant-NGP | 0.2641 | 0.7318 | 0.4976 | 0.555 | 0.208 | 27.976 | 0.799 | 0.255 |
| Ours | 0.0156 | **0.0197** | **0.0177** | **0.933** | **0.960** | **36.503** | **0.966** | **0.048** |

Specifically, in terms of rendering metrics, our method achieves a 5 dB higher rendering quality than methods like *Instant-NGP* and *DVGO*. Compared to the volume rendering-based reconstruction methods, our method achieves better view rendering performance. Furthermore, our method outperforms reconstruction-focused methods in these metrics. These further show the robustness of the proposed method on synthetic datasets. The qualitative results can be found in the Supplementary.

**Evaluation on Scannet Dataset.** We present the quantitative results (table 3) and the qualitative results (fig. 5 and fig. 6) evaluated on five real-world scenes. It can be seen from table 3 that our method achieves the best metrics of PSNR and Chamfer-$l_1$.

For the 3D reconstruction task, our approach yields smoother surface and more complete objects, *e.g.* whiteboard and TV (fig. 6). Remarkably, for the view synthesis task, our method successfully generates clear floor stripes and poster text, which appear blurred in other results (fig. 5).

**Table 2: Reconstruction and view synthesis results of Replica dataset. The best performances are highlighted in bold.**

| Method | Acc ↓ | Com ↓ | C-$l_1$ ↓ | NC ↑ | F-score ↑ | PSNR ↑ | SSIM ↑ | LPIPS ↓ |
|---|---|---|---|---|---|---|---|---|
| BundleFusion | 0.0145 | 0.0453 | 0.0299 | 0.961 | 0.936 | - | - | - |
| Neus | 0.1623 | 0.2956 | 0.2288 | 0.754 | 0.194 | 28.939 | 0.855 | 0.181 |
| VolSDF | 0.1348 | 0.3009 | 0.2180 | 0.747 | 0.339 | 30.375 | 0.866 | 0.175 |
| Neuralangelo | 0.4747 | 0.6288 | 0.5519 | 0.649 | 0.112 | 30.213 | 0.868 | 0.178 |
| Neural RGB-D | **0.0096** | 0.2447 | 0.1271 | 0.934 | 0.847 | 32.668 | 0.893 | 0.198 |
| Go-Surf | 0.0120 | 0.0122 | 0.0121 | 0.9718 | 0.9896 | 30.967 | 0.884 | 0.217 |
| DVGO | 0.2399 | 0.3511 | 0.2955 | 0.6040 | 0.239 | 31.962 | 0.893 | 0.223 |
| Instant-NGP | 0.3332 | 1.1151 | 0.7288 | 0.5470 | 0.1583 | 32.352 | 0.884 | 0.150 |
| Ours | 0.0112 | **0.0111** | **0.0112** | **0.9748** | **0.9911** | **37.104** | **0.955** | **0.074** |

**Table 3: Performance comparison with other nerf-based methods on rendering and reconstruction results of the Scannet dataset. Best results are highlighted as bold.**

| Method | scene0000 C-$l_1$ ↓ | scene0000 PSNR↑ | scene0002 C-$l_1$ ↓ | scene0002 PSNR↑ | scene0005 C-$l_1$ ↓ | scene0005 PSNR↑ | scene0024 C-$l_1$ ↓ | scene0024 PSNR↑ | scene0494 C-$l_1$ ↓ | scene0494 PSNR↑ | Average C-$l_1$ ↓ | Average PSNR↑ |
|---|---|---|---|---|---|---|---|---|---|---|---|---|
| Neus | 0.410 | 24.056 | 0.364 | 21.397 | 0.444 | 25.985 | 0.568 | 21.888 | 0.543 | 28.770 | 0.466 | 24.419 |
| VolSDF | 0.652 | 24.849 | 0.606 | 21.284 | 0.573 | 25.456 | 0.793 | 22.560 | 0.541 | 27.831 | 0.633 | 24.396 |
| Neuralangelo | 0.457 | 22.847 | 0.350 | 22.227 | 0.339 | 28.075 | 0.462 | 20.496 | 0.336 | 28.865 | 0.389 | 24.502 |
| Neural RGB-D | 0.068 | 21.779 | 0.060 | 17.330 | 0.075 | 23.635 | 0.185 | 16.266 | **0.044** | 28.274 | 0.086 | 21.457 |
| Go-surf | 0.030 | 24.033 | 0.024 | 20.246 | 0.054 | 24.977 | **0.106** | 21.538 | 0.103 | 27.572 | 0.063 | 23.673 |
| DVGO | 0.133 | 26.315 | 0.143 | 21.206 | 0.153 | 28.126 | 0.142 | 22.689 | 0.155 | 29.655 | 0.145 | 25.598 |
| Instant-NGP | 0.476 | 23.145 | 0.249 | 22.854 | 0.349 | 23.988 | 0.958 | 15.285 | 0.305 | 28.786 | 0.467 | 22.812 |
| **Ours** | **0.029** | **26.712** | **0.022** | **23.032** | **0.053** | **29.146** | **0.106** | **24.240** | 0.075 | **31.675** | **0.058** | **26.961** |

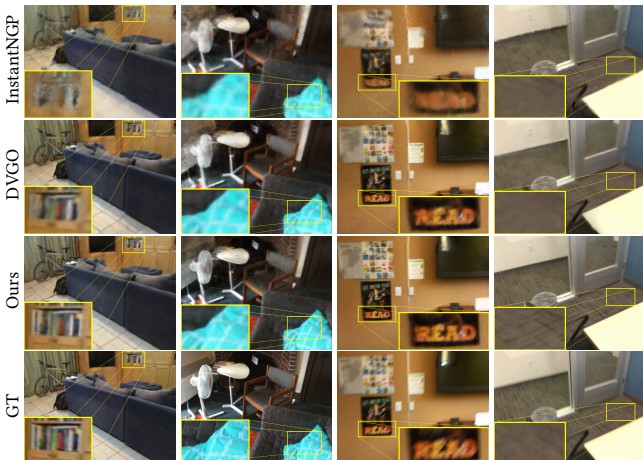

**Figure 5: Qualitative results on ScanNet scenes demonstrate the superior rendering quality of our approach compared to previous NeRF-based methods, especially for images exhibiting severe motion blur, such as the text on a poster (column 3) and the stripes on the floor (column 4).**

## 4.3 Ablations

We conduct ablation studies to demonstrate the effectiveness of designing the blocks and choice of scene representation. The quantitative result is shown in tables 4 to 7, where we evaluate these methods in the RGBD synthetic dataset. Other ablation studies can be found in the Supplementary.

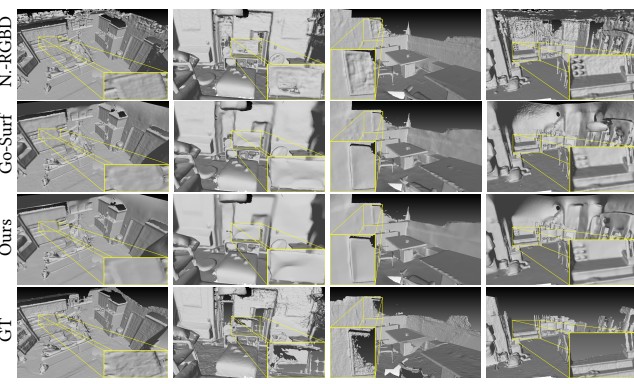

**Figure 6: Qualitative mesh reconstruction on Scannet scenes. Our method produces visually smoother and cleaner meshes compared to previous methods, as demonstrated by the zoomed-in details provided for comparison.**

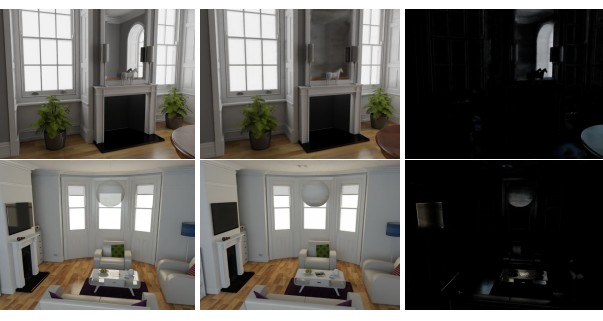

**(a) Complete color**     **(b) Vi color**     **(c) Vd color**

**Figure 7: Color decoupling results of the proposed method on NeuralRGBD dataset. The Vi color represents the view-independent color, while Vd color is the view-dependent color. It can be seen that our method can effectively decouple the complete color into the view-dependent (specular reflective surfaces) and view-independent (diffuse surfaces) colors.**

**Effect of the Du-NeRF architecture.** To analyze the network architecture of our dual-field, we experimented with various architectural configurations, detailed in table 4. The SDF-only represents that the method contains only the SDF branch and the Density-only involves only the density branch. Dual is the proposed method which has the density branch and SDF branch. The experimental results show that the two branches achieve optimal rendering and reconstruction performance. The SDF branch exhibits superior reconstruction capabilities, while the Density branch excels in rendering, as evidenced by the fact that the SDF-only branch outperforms the Density-only branch in terms of F-score, and the latter achieves higher PSNR values.

**Effect of color disentanglement.** To explore the impact of color decoupling on view rendering and 3D reconstruction, we conducted several tests summarized in table 5. FC indicates supervision with color loss without decoupling, while VdC+ViC involves color loss from both View-dependent Color (VdC) and View-independent Color (ViC), calculated separately in VdC and ViC conditions. The

**Table 4: Ablation study of the Du-NeRF architecture.**

| Method | C-$l_1$ ↓ | F-score ↑ | PSNR ↑ | LPIPS ↓ |
|---|---|---|---|---|
| SDF-only | 0.0133 | 0.974 | 31.741 | 0.110 |
| Density-only | 0.0168 | 0.934 | 32.744 | 0.090 |
| Dual(SDF+Density) | **0.0115** | **0.977** | **34.775** | **0.074** |

results presented in table 5 show that ViC supervision provides optimal performance for 3D reconstruction and volume rendering tasks. Furthermore, the comparable performance observed in the first two rows indicates that color decoupling does not improve reconstruction or rendering quality. In particular, the use of VdC as the supervision signal leads to a significant degradation in reconstruction and rendering performance compared to FC and VdC+ViC. fig. 7 demonstrates that our method can accurately decompose the ViC and VdC color. For more experimental results on color disentanglement, please refer to the Supplementary.

**Table 5: Ablation study of color disentanglement (C-D), where ViC (Ours) denotes the view-independent component, VdC is the view-dependent component, and FC represents the complete color, respectively.**

| C-D | Supervised | C-$l_1$ ↓ | F-score ↑ | PSNR ↑ | LPIPS ↓ |
|---|---|---|---|---|---|
| w/o | FC | 0.0182 | 0.9549 | 34.494 | 0.090 |
| w/ | VdC+ViC | 0.0185 | 0.9526 | 34.710 | 0.080 |
| w/ | VdC | 0.1658 | 0.6503 | 32.780 | 0.122 |
| w/ | ViC | **0.0177** | **0.9597** | **35.225** | **0.072** |

**Effect of different grid representation.** We examined how different scene representations affect reconstruction and rendering using three strategies, detailed in table 6. $GM − CM$ uses four-level resolution grids for both geometry and color, $GH − CH$ employs hash-based grids for both, and $GM − CH$ combines a four-level grid for geometry with a hash-based grid for color. Findings in table 6 indicate that while hash-based color grids improve rendering outcomes, hash-based geometry grids reduce reconstruction quality. The mixed representation of $GM−CH$ yields the best results in both F-score and PSNR, suggesting that color benefits from a complex representation, whereas geometry performs better with a simpler way.

**Table 6: Ablation study of different scene representation, where $G_M, C_M, G_H, C_H$, denote geometric multi-resolution grid, color multi-resolution grid, geometric hash grid, and color hash grid, respectively.**

| Method | C-$l_1$ ↓ | F-score ↑ | PSNR ↑ | LPIPS ↓ |
|---|---|---|---|---|
| $G_M − C_M$ | **0.0177** | 0.9597 | 35.225 | 0.072 |
| $G_H − C_H$ | 0.0294 | 0.8976 | 36.087 | 0.108 |
| $G_M − C_H$ | **0.0177** | **0.9600** | **36.503** | **0.048** |

**Effect of pose refinement and PFA evaluation strategy.** To assess the impact of pose optimization on 3D reconstruction, we tested two scenarios: with and without pose optimization, as illustrated in fig. 8. Results indicate that lacking pose optimization

leads to defective meshes with unwanted floaters and compromised geometry.

Further, to ascertain the correct poses for testing images, we explored three configurations. "W/o Alignment" uses uncorrected provided poses, "IA" (Interpolated Alignment) derives poses from averaging those of two adjacent training images, and "PFA" (Proximity Frame Alignment) is our method. Results in table 7 show that without proper pose alignment, view rendering quality significantly declines. Notably, our PFA method surpasses the IA strategy, which relies on simple interpolation, in rendering quality.

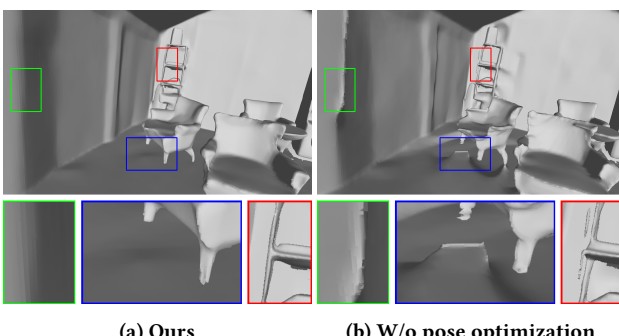

(a) Ours          (b) W/o pose optimization

**Figure 8: The comparison experiments about pose optimization. Floating points and artifacts are present in the scene when pose optimization is turned off.**

**Table 7: Evaluation of pose alignment strategy, included W/o Alignment, Interpolation Alignment (IA) and Proximity Frame Alignment (PFA). It can be seen that our PFA strategy could achieve the best performance.**

| Method | PSNR ↑ | SSIM ↑ | LPIPS ↓ |
|---|---|---|---|
| W/o Alignment | 18.357 | 0.679 | 0.398 |
| IA | 21.802 | 0.674 | 0.342 |
| PFA (ours) | **26.712** | **0.796** | **0.296** |

## 5 CONCLUDING REMARKS

We have presented a dual neural radiance field (Du-NeRF) for high-fidelity 3D indoor scene reconstruction and rendering simultaneously. This method incorporates two branches, one for reconstruction and another for rendering, allowing mutual enhancement. Additionally, we developed an effective framework that facilitates the on-the-fly extraction of view-independent color information from the model, which can be leveraged for supervising 3D reconstruction, potentially yielding smoother and more complete mesh representations.

Besides, the effectiveness of the proposed method remains to be tested under a few-shot scenario where a small number of RGBD images are provided. In addition, trying new feature representations such as replacing hash representation with Gaussian splitting may improve performances. We will address these in future works.

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
