# OpenReview forum: "3D Reconstruction and Novel View Synthesis of Indoor Environments based on a Dual Neural Radiance Field"
_acmmm.org/ACMMM/2024/Conference — MM2024 Poster_

### Official Review · Reviewer_95DV · 2024-06-08

**Rating:** 5
**Confidence:** 3

**Summary:**

This paper propsoes a framework, Du-NeRF, for novel view synthesis and 3D reconstruction for indoor environements. For estimating accurate geometric information of a scene, Du-NeRF utilizes SDF and density field information. The authors suggest that their method enhances both reconstruction and rendering performance.

**Strengths:**

The authors have done extensive experiments to validate the performance of their method in 3D reconstruction task. Furthermore, they have proposed a novel self-supervised method to extract a view-independent color components to enhance the quality of 3D reconstruction. Through using both density field and SDF field, the proposed method seems to outperform previous works in recovering details in the 3D scene and generating novel-view.

**Limitations:**

The images presented to show qualitative results are small. Increasing the size of images in qualitative result could help readers to easily compare the results.
In addition, the overall architecture shown in Figure 2 lacks labeling. It is difficult for readers to understand the architecture immediately.

**Suitability:**

2

---

### Official Review · Reviewer_9nTG · 2024-06-08

**Rating:** 3
**Confidence:** 3

**Summary:**

In this paper, authors proposed Du-NeRF for novel view synthesis and 3D reconstruction for indoor environments. In Du-NeRF, it is using  SDF field, and the density fields to extract geometric information of the scene. In addition the authors applied self-supervision method to enhance the quality of 3D reconstruction.

**Strengths:**

The authors have done extensive experiments to validate their method's performance in 3D reconstruction and novel-view synthesis.
The ablation studies are well organized based on its architecture.
They have analyzed their methods enough for readers to understand the effects of each proposed module/methods.

**Limitations:**

Disentangling SDF and Density field may be novel, but use of these two information is a very commonly used methods when implementing NeRF framework. Use of these information could be easily found in previous works. Hence, the contribution of using both fields lacks novelty.

**Suitability:**

2

---

### Official Review · Reviewer_YSzX · 2024-06-08

**Rating:** 5
**Confidence:** 3

**Summary:**

This paper is suggesting Du-NeRF for novel view synthesis and 3D reconstruction for indoor environments. In Du-NeRF, it is using 2 geometiric fields, one consists of SDF field, and the other consists of density fields. Also, it is using self-supervised learning for extracting view-independent color component. By using Du-NeRF, the authors propose that it can enhance both reconstruction and rendering performance. Many experiments and ablation studies prove its novelty.

**Strengths:**

The authors proposed Du-NeRF, which focus on both SDF and Density. By using Dual NeRF and focusing on each part, it is both improving 3D reconstruction and new view synthesis. It is proved by Table1 of experiments.

Also, the authors proposed self-supervised method for more high quality of getting view-independent color component in 3D reconstruction. By using feature from SDF branch and Density branch, using self-supervised learning in appearance branch shows novelty on extracting color.

Experiments and ablation studies are conducted adequately. Especially, ablation studies are well organized based on its architecture.

**Limitations:**

In figure 2, it does not state about the color components (e.g., final color C, color components cdi, csi). Explicitly including variables in figure would be more helpful to understand at a glance.

**Suitability:**

2

---

### Meta-Review · Area_Chair_mvr1 · 2024-07-01

**Recommendation:** Accept (Poster)
**Confidence:** 4

**Metareview:**

The paper proposes a method which utilizes two different types of geometric field which allows accurate reconstruction of a 3D object. The utilized SDF field and density field are commonly used in this field of studies. It is true that the disentanglement of two field to fully use the information contained in these two fields. However, constituting two commonly used fields seems to lack novelty for publication.


***TPC Addendum***
The paper received a split in recommendations from the reviewers and the AC. Given the importance of the topic, high average scores, and trend toward score increase after rebuttal, the TPC suggests that the debate/discussion continue at the conference with a poster presentation.